# Natural Graph Networks

**Pim de Haan**
Qualcomm AI Research*
University of Amsterdam QUVA Lab

**Taco Cohen**
Qualcomm AI Research

**Max Welling**
Qualcomm AI Research
University of Amsterdam

## Abstract

A key requirement for graph neural networks is that they must process a graph in a way that does not depend on how the graph is described. Traditionally this has been taken to mean that a graph network must be equivariant to node permutations. Here we show that instead of equivariance, the more general concept of naturality is sufficient for a graph network to be well-defined, opening up a larger class of graph networks. We define global and local natural graph networks, the latter of which are as scalable as conventional message passing graph neural networks while being more flexible. We give one practical instantiation of a natural network on graphs which uses an equivariant message network parameterization, yielding good performance on several benchmarks.

## 1 Introduction

Graph-structured data is among the most ubiquitous forms of structured data used in machine learning and efficient practical neural network algorithms for processing such data have recently received much attention [Wu et al., 2020]. Because of their scalability to large graphs, graph convolutional neural networks or message passing networks are widely used. However, it has been shown [Xu et al., 2018] that such networks, which pass messages along the edges of the graph and aggregate them in a permutation invariant manner, are fundamentally limited in their expressivity.

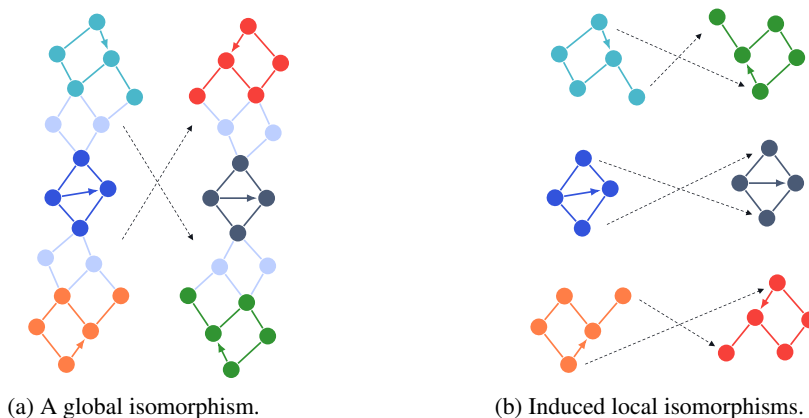

(a) A global isomorphism.      (b) Induced local isomorphisms.

Figure 1: A global graph isomorphism corresponds for each edge to a local isomorphism on its neighbourhood, shown for three example edges - denoted with arrows. Hence, when a message passing kernel satisfies the naturality condition for local isomorphisms of the edge neighbourhood (Eq. 4), it also satisfies the global naturality condition (Eq. 2).

More expressive equivariant graph networks exist [Maron et al., 2018], but these treat the entire graph as a monolithic linear structure (e.g. adjacency matrix) and as a result their computational cost scales superlinearly with the size of the graph. In this paper we ask the question: how can we design maximally expressive graph networks that are equivariant to *global* node permutations while using only *local* computations?

If we restrict a global node relabeling / permutation to a local neighbourhood, we obtain a graph isomorphism between local neighbourhoods (see Figure 1). If a locally connected network is to be equivariant to global node relabelings, the message passing scheme should thus process isomorphic neighbourhoods in an identical manner. Concretely, this means that weights must be shared between isomorphic neighbourhoods. Moreover, when a neighbourhood is symmetrical (Figure 1), it is isomorphic to itself in a non-trivial manner, and so the convolution kernel has to satisfy an equivariance constraint with respect to the symmetry group of the neighbourhood.

Local equivariance has previously been used in gauge equivariant neural networks [Cohen et al., 2019]. However, as the local symmetries of a graph are different on different edges, we do not have a single gauge group here. Instead, we have more general structures that can be captured by elementary category theory. We thus present a categorical framework we call *natural graph networks* that can be used describe maximally flexible global and local graph networks. In this framework, an equivariant kernel is "just" a natural transformation between two functors. We will not assume knowledge of category theory in this paper, and explicit category theory is limited to Section 5.

When natural graph networks (NGNs) are applied to graphs that are regular lattices, such as a 2D square grid, or to a highly symmetrical grid on the icosahedron, one recovers conventional equivariant convolutional neural networks [Cohen and Welling, 2016, Cohen et al., 2019]. However, when applied to irregular grids, like knowledge graphs, which generally have few symmetries, the derived kernel constraints themselves lead to impractically little weight sharing. We address this by parameterizing the kernel with a message network, an equivariant graph network which takes as input the local graph structure. We show that our kernel constraints coincide with the constraints on the message network being equivariant to node relabelings, making this construction universal whenever the network that parameterizes the kernel is universal.

## 2 Global Natural Graph Networks

As mentioned before, there are many equivalent ways to encode (directed or undirected) graphs. The most common encoding used in the graph neural networks literature is to encode a graph as a (node-node) adjacency matrix $A$, whose rows and columns correspond to the nodes and whose $(i, j)$-th entry signals the presence ($A_{ij} = 1$) or absence ($A_{ij} = 0$) of an edge between node $i$ and $j$. There are many other options, but here we will adopt the following definition:

**Definition 2.1.** A *Concrete Graph* $G$ is a finite set of nodes[2] $\mathcal{V}(G) \subset \mathbb{N}$ and a set of edges $\mathcal{E}(G) \subset \mathcal{V}(G) \times \mathcal{V}(G)$.

The natural number labels of the nodes of a concrete graph are essential for representing a graph in a computer, but contain no actual information about the underlying graph. Hence, different concrete graphs that are related by a relabelling, encode the graphs that are essentially the same. Such relabellings are called graph isomorphisms.

**Definition 2.2** (Graph isomorphism and automorphism)**.** Let $G$ and $G'$ be two graphs. An isomorphism $\phi : G \to G'$ is a mapping (denoted by the same symbol) $\phi : \mathcal{V}(G) \to \mathcal{V}(G')$ that is bijective and preserves edges, i.e. satisfies for all $(i, j) \in \mathcal{V}(G) \times \mathcal{V}(G)$:

$$(i, j) \in \mathcal{E}(G) \iff (\phi(i), \phi(j)) \in \mathcal{E}(G'). \tag{1}$$

If there exists an isomorphism between $G$ and $G'$, we say they are *isomorphic*. An isomorphism from a graph to itself is also known as an *automorphism* or simply symmetry.

In order to define graph networks, we must first define the vector space of features on a graph. Additionally, we need to define how the feature spaces of isomorphic graphs are related, so we can express a feature on one concrete graph on other isomorphic concrete graphs.

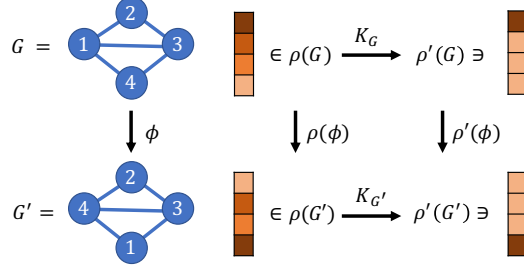

Figure 2: A graph feature $\rho$ assigns to each graph $G$ a vector space $\rho(G)$ (here $\rho(G) = \rho(G') = \mathbb{R}^4, \rho = \rho'$) and to each graph isomorphism $\phi : G \to G'$ a linear map $\rho(\phi) : \rho(G) \to \rho(G')$ (here swapping the first and fourth row). Global Natural Graph Network layer $K$ between features $\rho$ and $\rho'$ has for each graph $G$ a map $K_G : \rho(G) \to \rho'(G)$, such that for each graph isomorphism $\phi : G \to G'$ the above naturality diagram commutes.

**Definition 2.3** (Graph feature space)**.** A graph feature space, or graph representation, $\rho$ associates to each graph $G$ a vector space $V_G = \rho(G)$, and to each graph isomorphism $\phi : G \to G'$ an invertible linear map $\rho(\phi) : V_G \to V_{G'}$, such that the linear maps respect composition of graph isomorphisms: $\rho(\phi \circ \phi') = \rho(\phi) \circ \rho(\phi')$. [3]

As the nodes in a concrete graph have a unique natural number as a label, the nodes can be ordered. A graph isomorphism $\phi : G \to G'$ induces a permutation of that ordering. This gives a convenient way of constructing graph feature spaces. For example, for the vector representation, we associate with graph $G$ the vector space $\rho(G) = \mathbb{R}^{|\mathcal{V}(G)|}$ and associate to graph isomorphisms the permutation matrix of the corresponding permutation. Similarly, for the matrix representation, we associate to graph $G$ feature matrix vector space $\rho(G) = \mathbb{R}^{|\mathcal{V}(G)| \times |\mathcal{V}(G)|}$ and to graph isomorphism $\phi : G \to G'$, linear map $\rho(\phi)(v) = PvP^T$, where $P$ is the permutation matrix corresponding to $\phi$.

A neural network operating on such graph features can, in general, operate differently on different graphs. Its (linear) layers, mapping from graph feature space $\rho$ to feature space $\rho'$, thus has for each possible graph $G$, a (linear) map $K_G : \rho(G) \to \rho'(G)$. However, as isomorphic graphs $G$ and $G'$ are essentially the same, we will want $K_G$ and $K_{G'}$ to process the feature space in an equivalent manner.

**Definition 2.4** (Global Natural Graph Network Layer)**.** A layer (or linear layer) in a global natural graph network (GNGN) is for each concrete $G$ a map (resp. linear map) $K_G : \rho(G) \to \rho'(G)$ between the input and output feature spaces such that for every graph isomorphism $\phi : G \to G'$, the following condition ("naturality") holds:

$$\rho'(\phi) \circ K_G = K_{G'} \circ \rho(\phi). \tag{2}$$

Equivalently, the following diagram should commute:

$$
\begin{array}{ccc}
\rho(G) & \xrightarrow{K_G} & \rho'(G) \\
\downarrow{\scriptstyle\rho(\phi)} & & \downarrow{\scriptstyle\rho'(\phi)} \\
\rho(G') & \xrightarrow{K_{G'}} & \rho'(G')
\end{array}
$$

The constraint on the layer (Eq. 2) says that if we first transition from the input feature space $\rho(G)$ to the equivalent input feature space $\rho(G')$ via $\rho(\phi)$ and then apply $K_{G'}$ we get the same thing as first applying $K_G$ and then transitioning from the output feature space $\rho'(G)$ to $\rho'(G')$ via $\rho'(\phi)$. Since $\rho(\phi)$ is invertible, if we choose $K_G$ for some $G$ then we have determined $K_{G'}$ for any isomorphic $G'$ by $K_{G'} = \rho'(\phi) \circ K_G \circ \rho(\phi)^{-1}$. Moreover, for any automorphism $\phi : G \to G$, we get a equivariance constraint $\rho'(\phi) \circ K_G = K_G \circ \rho(\phi)$. Thus, to choose a layer we must choose for each isomorphism class of graphs one map $K_G$ that is equivariant to automorphisms. For linear layers, these can in principle be learned by first finding a complete solution basis to the automorphism equivariance constraint, then linearly combining the solutions with learnable parameters.

The construction of the graph isomorphisms, the graph feature space and the natural graph network layer resemble mathematical formalization that are used widely in machine learning: groups, group representations and equivariant maps between group representations. However, the fact that the natural graph network layer can be different for each graph, suggests a different formalism is needed, namely the much more general concepts of a category, a functor and a natural transformation. How natural transformations generalize over equivariant maps is described in section 5.

## 2.1 Relation to Equivariant Graph Networks

The GNGN is a generalization of equivariant graph networks (EGN) [Maron et al., 2018, 2019], as an EGN can be viewed as a GNGN with a particular choice of graph feature spaces and layers. The feature space of an EGN for a graph of $n$ nodes is defined by picking a group representation of the permutation group $S_n$ over $n$ symbols. Such a representation consists of a vector space $V_n$ and an invertible linear map $\rho(\sigma) : V_n \to V_n$ for each permutation $\sigma \in S_n$, such that $\rho(\sigma\sigma') = \rho(\sigma) \circ \rho(\sigma')$. A typical example is $V_n = \mathbb{R}^{n \times n}$, with $\rho(\sigma)$ acting by permuting the rows and columns. The (linear) layers of an EGN between features $\rho$ and $\rho'$ are (linear) maps $K_n : V_n \to V'_n$, for each $n$, such that the map is equivariant: $\rho'(\sigma) \circ K_n = K_n \circ \rho(\sigma)$ for each permutation $\sigma \in S_n$.

Comparing the definitions of EGN features and layers to GNGN features and layers, we note the former are instances of the latter, but with the restriction that an EGN picks a single representation vector space $V_n$ and single equivariant map $K_n$ for all graphs of $n$ nodes, while in a general GNGN, the representation vector space and equivariant map can arbitrarily differ between non-isomorphic graphs. In an EGN, the graph structure must be encoded as a graph feature. For example, the adjacency matrix can be encoded as a matrix representation of the permutation group. Such constructions are shown to be universal [Keriven and Peyré, 2019], but impose considerable constraints on the parameterization. For example, one may want to use a GNGN with completely separate sets of parameters for non-isomorphic graphs, which is impossible to express as an EGN.

# 3 Local Graph Networks

Global NGNs provide a general framework of specifying graph networks that process isomorphic graphs equivalently. However, in general, its layers perform global computations on entire graph features, which has high computational complexity for large graphs.

## 3.1 Local Invariant Graph Networks

An entirely different strategy to building neural networks on graphs is using graph convolutional neural networks or message passing networks [Kipf and Welling, 2016, Gilmer et al., 2017]. We will refer to this class of methods as local invariant graph networks (LIGNs). Such convolutional architectures are generally more computationally efficient compared to the global methods, as the computation cost of computing one linear transformation scales linearly with the number of edges.

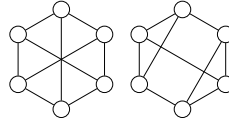

Figure 3: Two regular graphs.

LIGNs are instances of GNGNs, where the feature space for a graph consists of a copy of the same vector space $V_{\mathcal{N}}$ at each node, and graph isomorphisms permute these node vector spaces. In their simplest form, the linear layers of an LIGN pass messages along edges of the graph:

$$K_G(v)_p = \sum_{(p,q) \in \mathcal{E}} W v_q, \tag{3}$$

where $v_p \in \mathcal{V}_{\mathcal{N}}$ is a feature vector at node $p$ and $W : V_{\mathcal{N}} \to V'_{\mathcal{N}}$ is a single matrix used on each edge of any graph. This model can be generalized into using different aggregation functions than the sum and having the messages also depend on $v_p$ instead of just $v_q$ [Gilmer et al., 2017]. It is easy to see that these constructions satisfy the GNGN constraint (Eq. 2), but also result in the output $K_G(v)_p$ being *invariant* under a permutation of its neighbours, which is the reason for the limited expressivity noted by [Xu et al., 2018]. For example, no invariant message passing network can discriminate between the two regular graphs in figure 3. Furthermore, if applied to the rectangular pixel grid graph of an image, it corresponds to applying a convolution with isotropic filters.

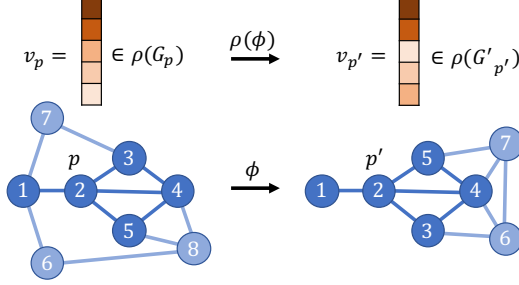

Figure 4: A node feature $\rho$ assigns to each node neighbourhood $G_p$ (here the dark colored nodes around node $p$) a vector space $\rho(G_p)$ (here $\rho(G_p) = \mathbb{R}^5$) and to each local node isomorphism $\psi : G_p \to G'_{p'}$ a linear map $\rho(\psi) : \rho(G) \to \rho(G')$ (here swapping the third and fifth row).

## 3.2   Local Natural Graph Networks

The idea of a Local Natural Graph Network (LNGN) is to implement a scalable GNGN layer that consists of passing messages along edges with a message passing kernel and then aggregating the incoming messages. It generalises over local invariant graph networks by making the node features transform under isomorphisms of the neighbourhood of the node and by allowing different message passing kernels on non-isomorphic edges.

**Definition 3.1** (Neighbourhoods and local isomorphisms). A *node neighbourhood*[4] $G_p$ is a subgraph $G_p$ of a concrete graph $G$ in which one node $p \in \mathcal{V}(G_p)$ is marked. Subgraph $G_p$ inherits the node labels from $G$, making $G_p$ a concrete graph itself. A *local node isomorphism* is a map between node neighbourhoods $\psi : G_p \to G'_{p'}$, consisting of a graph isomorphism $\psi : G_p \to G'_{p'}$ such that $\psi(p) = p'$. Similarly, an *edge neighbourhood* is a concrete graph $G_{pq}$ with a marked edge $(p, q)$ and a *local edge isomorphism* that maps between edge neighbourhoods such that the marked edge is mapped to the marked edge.

Given a graph $G$, we can assign to node $p \in \mathcal{V}(G)$ a node neighbourhood $G_p$ in several ways. In our experiments, we choose $G_p$ to contain all nodes in $G$ that are at most $k$ edges removed from $p$, for some natural number $k$, and all edges between these nodes. Similarly, we pick for edge $(p, q) \in \mathcal{E}(G)$ neighbourhood $G_{pq}$ containing all nodes at most $k$ edges removed from $p$ or $q$ and all edges between these nodes. In all experiments, we chose $k = 1$, unless otherwise noted. General criteria for the selection of neighbourhoods are given in App. C. Neighbourhood selections satisfying these criteria have that any global graph isomorphism $\phi : G \to G'$, when restricted to a node neighbourhood $G_p$ equals a node isomorphism $\phi_p : G_p \to G'_{p'}$ and when restricted to an edge neighbourhood $G_{pq}$ equals a local edge isomorphism $\phi_{pq} : G_{pq} \to G'_{p'q'}$. Furthermore, it has as a property that any local edge isomorphism $\psi : G_{pq} \to G'_{p'q'}$ can be restricted to node isomorphisms $\psi_p : G_p \to G'_{p'}$ and $\psi_q : G_q \to G'_{q'}$ of the start and tail node of the edge.

Next, we choose a feature space for the local NGN by picking a node feature space $\rho$, which is a graph feature space (Def. 2.4) for node neighbourhoods in complete analogy with the previous section on global NGNs. Node feature space $\rho$ consists of selecting for any node neighbourhood $G_p$ a vector space $\rho(G_p)$ and for any local node isomorphism $\phi : G_p \to G'_{p'}$, a linear bijection $\rho(\phi) : \rho(G_p) \to \rho(G'_{p'})$, respecting composition: $\rho(\phi) \circ \rho(\phi') = \rho(\phi \circ \phi')$.

A node neighbourhood feature space $\rho$ defines a graph feature space $\hat{\rho}$ on global graphs by concatenating (taking the direct sum of) the node vector spaces: $\hat{\rho}(G) = \bigoplus_{p \in \mathcal{V}(G)} \rho(G_p)$. For a global feature vector $v \in \hat{\rho}(G)$, we denote for node $p \in \mathcal{V}(G)$ the feature vector as $v_p \in \rho(G_p)$. The global graph feature space assigns to global graph isomorphism $\phi : G \to G'$ a linear map $\hat{\rho}(\phi) : \hat{\rho}(G) \to \hat{\rho}(G')$, which permutes the nodes and applies $\rho$ to the individual node features:

$$\hat{\rho}(\phi)(v)_{\phi(p)} = \rho(\phi_p)(v_p)$$

Given two such node feature spaces $\rho$ and $\rho'$, we can define a (linear) local NGN message passing kernel $k$ by choosing for each possible edge neighbourhood $G_{pq}$ a (linear) map $k_{pq} : \rho(G_p) \to \rho'(G_q)$,

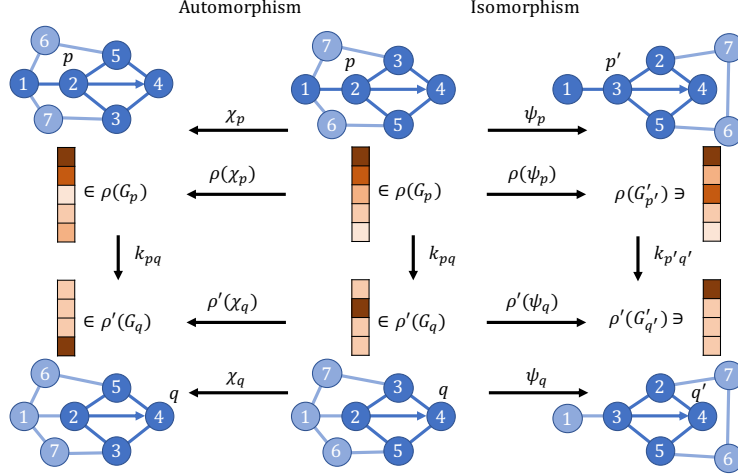

Figure 5: Local Natural Graph Network kernel $k$ between node features $\rho$ and $\rho'$ consists of a map $k_{pq}$ : $\rho(G_p) \to \rho'(G_q)$ for each edge $(p, q)$, satisfying the above commuting diagrams for each edge isomorphism $\psi : G_{pq} \to G'_{p'q'}$ and automorphism $\chi : G_{pq} \to G_{pq}$. In this example, the node neighbourhoods of $p, p', q$ and $q'$ are colored dark. Edge isomorphism $\psi$, which swaps nodes 1 and 5, restricts to node isomorphisms $\psi_p$ and $\psi_q$ on input and output node neighbourhoods. The associated linear maps $\rho(\psi_p)$ and $\rho'(\psi_q)$ swap second and third row and first and second row respectively - corresponding to the reordering of the nodes in the neighbourhood by the node isomorphism. Similarly, the automorphism $\chi$ swaps nodes 3 and 5. The isomorphism leads to weight sharing between $k_{pq}$ and $k_{p'q'}$ and the automorphism to a kernel constraint on $k_{pq}$.

which takes the role of $W$ in Eq. 3. These maps should satisfy that for any edge neighbourhood isomorphism $\psi : G_{pq} \to G'_{p'q'}$, we have that

$$\rho'(\psi_q) \circ k_{pq} = k_{p'q'} \circ \rho(\psi_p). \tag{4}$$

In words, this "local naturality" criterion states that passing the message along an edge from $p$ to $q$, then transporting with a local isomorphism to $q'$ yields the same result as first transporting from $p$ to $p'$, then passing the message along the edge to $q'$. In analogy to the global NGN layer, we have that isomorphisms between different edge neighbourhoods bring about weight sharing - with a change of basis given by Eq. 4, while automorphisms create constraints on the kernel $k$.

Using the local NGN kernel $k$ between node feature spaces $\rho$ and $\rho'$, we can define a global NGN layer between graph feature spaces $\hat{\rho}$ and $\hat{\rho}'$ as:

$$K_G(v)_q = \sum_{(p,q) \in \mathcal{E}(G)} k_{pq}(v_p) \tag{5}$$

The following main result, proven in Appendix D, shows that this gives a global NGN layer.

**Theorem 1.** *Let $k$ be a local NGN kernel between node feature spaces $\rho$ and $\rho'$. Then the layer in equation 5 defines a global NGN layer between the global graph feature spaces $\hat{\rho}$ and $\hat{\rho}'$, satisfying the global NGN naturality condition (Eq. 2).*

In appendix F, we show when a local NGN is applied to a regular lattice, which is a graph with a global transitive symmetry, the NGN is equivalent to a group equivariant convolutional neural network [Cohen and Welling, 2016], when the feature spaces and neighbourhoods are chosen appropriately. In particular, when the graph is a square grid with edges on the diagonals, we recover an equivariant planar CNN with 3x3 kernels. Bigger kernels are achieved by adding more edges. When the graph is a grid on a locally flat manifold, such as a icosahedron or another platonic solid, and the grid is a regular lattice, except at some corner points, the NGN is equivalent to a gauge equivariant CNN [Cohen et al., 2019], except around the corners.

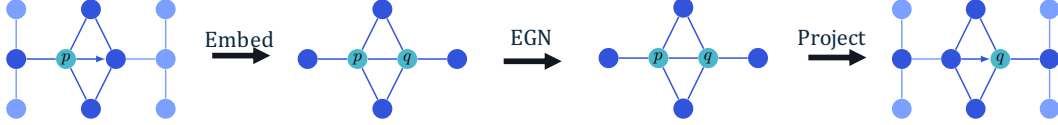

Figure 6: Local NGN message passing with an equivariant graph network kernel. The node feature $v_p$ at $p$ can be embedded into a graph feature $v_{p \to q}$ of the edge neighbourhood, to which any equivariant graph neural network can be applied. The output graph feature $v'_{p \to q}$ can be projected to obtain the message from $p$ to $q$, $v'^p_q$. The messages to $q$ are invariantly aggregated to form output feature $v'_q$.

## 4  Graph Neural Network Message Parameterization

Local naturality requires weight sharing only between edges with isomorphic neighbourhoods, so, in theory, one can use separate parameters for each isomorphism class of edge neighbourhoods to parameterize the space of natural kernels. In practice, graphs such as social graphs are quite heterogeneous, so that that few edges are isomorphic and few weights need to be shared, making learning and generalization difficult. This can be addressed by re-interpreting the message from $p$ to $q$, $k_{pq} v_p$, as a function $k(G_{pq}, v_p)$ of the edge neighbourhood $G_{pq}$ and feature value $v_p$ at $p$, potentially generalized to being non-linear in $v_p$, and then letting $k$ be a neural network-based "message network".

Local naturality (Eq. 4) can be guaranteed, even without explicitly solving kernel constraints for each edge in the following way. By construction of the neighbourhoods, the node feature $v_p$ can always be embedded into an edge feature, a graph feature $v_{p \to q}$ of the edge neighbourhood $G_{pq}$. The resulting graph feature can then be processed by an appropriate equivariant graph neural network operating on $G_{pq}$, in which nodes $p$ and $q$ have been distinctly marked, e.g. by a additional feature. The output graph feature $v'_{p \to q}$ can be restricted to create a node feature $v'^p_q$ at $q$, which is the message output. The messages are then aggregated using e.g. summing to create the convolution output $v'_q = \sum_{(p,q) \in \mathcal{E}} v'^p_q$. This is illustrated in figure 6. It is proven in appendix E that the graph equivariance constraint on the message network ensures that the resulting message satisfies the local naturality constraint (Eq. 4).

The selection of the type of graph feature and message network forms a large design space of natural graph networks. If, as in the example above, the node feature $v_p$ is a vector representation of the permutation of the node neighbourhood, the feature can be embedded into an invariant scalar feature of the edge neighbourhood graph by assigning an arbitrary node ordering to the edge neighbourhood and transporting from the node neighbourhood to the edge neighbourhood, setting a 0 for nodes outside the node neighbourhood. Any graph neural network with invariant features can subsequently be used to process the edge neighbourhood graph feature, whose output we restrict to obtain the message output at $q$. As a simplest example, we propose GCN$^2$, which uses an invariant message passing algorithm, or Graph Convolutional Neural Network [Kipf and Welling, 2016], on graph $\mathcal{G}_{pq}$ as message network.

## 5  Naturality as Generalization of Equivariance

As explained in Section 2.1, the difference between a global natural graph network and an equivariant graph network is that the GNGN does not require that non-isomorphic graphs are processed similarly, while the EGN requires all graphs to be processed the same. EGNs can be understood in terms of groups, representations and equivariant maps, but the more general GNGN requires the more general framework category theory, originally developed in algebraic topology, but recently also used as a modelling tool for more applied problems [Fong and Spivak, 2018]. Its constructions give rise to an elegant framework for building equivariant message passing networks, which we call "Natural Networks", potentially applicable beyond graph networks. In this section, we will outline the key ingredients of natural networks. We refer a reader interested in learning more about category theory to Leinster [2016] and Fong and Spivak [2018].

A (small) category $\mathcal{C}$ consists of a set of objects $\mathrm{Ob}(\mathcal{C})$ and for each two objects, $X, Y \in \mathrm{Ob}(\mathcal{C})$, a set of abstract (homo)morphisms, or arrows, $f \in \mathrm{Hom}_{\mathcal{C}}(X, Y)$, $f : X \to Y$ between them. The arrows can be composed associatively into new arrows and each object has an identity arrow $\mathrm{id}_X : X \to X$ with the obvious composition behaviour. When arrow $f : X \to Y, g : Y \to X$ compose to identities on $X$ and $Y$, they are isomorphisms (with $f^{-1} = g$).

A map between two categories $\mathcal{C}$ and $\mathcal{D}$ is a functor $F : \mathcal{C} \to \mathcal{D}$, when it maps each object $X \in \text{Ob}(\mathcal{C})$ to an object $F(X) \in \text{Ob}(\mathcal{D})$ and to each morphism $f : X \to Y$ in $\mathcal{C}$, a morphism $F(f) : F(X) \to F(Y)$ in $\mathcal{D}$, such that $F(g \circ f) = F(g) \circ F(f)$. Given two functors $F, G : \mathcal{C} \to \mathcal{D}$, a natural transformation $\eta : F \Rightarrow G$ consists of, for each object $X \in \text{Ob}(\mathcal{C})$, a morphism $\eta_X : F(X) \to F(Y)$, such that for each morphism $f : X \to Y$ in $\mathcal{C}$, the following diagram commutes, meaning that the two compositions $\eta_Y \circ F(f), G(f) \circ \eta_X : F(X) \to G(Y)$ are the same:

$$
\begin{array}{ccc}
F(X) & \xrightarrow{\eta_X} & G(X) \\
\downarrow{\scriptstyle F(f)} & & \downarrow{\scriptstyle G(f)} \\
F(Y) & \xrightarrow{\eta_Y} & G(Y)
\end{array}
\tag{6}
$$

A group is an example of a category with one object and in which all arrows, corresponding to group elements, are isomorphisms. Group representations are functors from this category to the category of vector spaces, mapping the single object to a vector space and morphisms to linear bijections of this space. The functor axioms specialise exactly to the axioms of a group representation. A natural transformation between such functors is exactly an equivariant map. As the group category has only one object, the natural transformation consists of a single morphism (linear map). Equivariant Graph Networks on graphs with $N$ nodes are examples of these, in which the group is the permutation group $S_N$, the representation space are $N \times N$ matrices, whose columns and rows are permuted by the group action, and the layer is a single equivariant map.

To study global NGNs, we define a category of graphs, whose objects are concrete graphs and morphisms are graph isomorphisms. The graph feature spaces (Def. 2.4) are functors from this graph category to the category Vec of vector spaces. The GNGN layer is a natural transformation between such functors, consisting of a different map for each graph, but with a naturality constraint (Eq. 6) for each graph isomorphism (including automorphisms).

Similarly, for local NGNs, we define a category $\mathcal{C}$ of node neighbourhoods and local node isomorphisms and a category $\mathcal{D}$ of edge neighbourhoods and local edge isomorphisms. A functor $F_0 : \mathcal{D} \to \mathcal{C}$ maps an edge neighbourhood to the node neighbourhood of the start node and an edge isomorphisms to the node isomorphism of the start node – which is well defined by the construction of the neighbourhoods. Similarly, functor $F_1 : \mathcal{D} \to \mathcal{C}$ maps to the neighbourhood of the tail node of the edge. Node feature spaces are functors $\rho, \rho' : \mathcal{C} \to \text{Vec}$. Composition of functors leads to two functors $\rho \circ F_0, \rho' \circ F_1 : \mathcal{D} \to \text{Vec}$, mapping an edge neighbourhood to the input feature at the start node or the output feature at the end node. A local NGN kernel $k$ is a natural transformation between these functors.

# 6 Related Work

As discussed above, prior graph neural networks can be broadly classified into local (message passing) and global equivariant networks. The former in particular has received a lot of attention, with early work by [Gori et al., 2005, Kipf and Welling, 2016]. Many variants have been proposed, with some influential ones including [Gilmer et al., 2017, Veličković et al., 2018, Li et al., 2017]. Global methods include [Hartford et al., 2018, Maron et al., 2018, 2019, Albooyeh et al., 2019]. We note that in addition to these methods, there are graph convolutional methods based on spectral rather than spatial techniques [Bruna et al., 2014, Defferrard et al., 2016, Perraudin et al., 2018].

Covariant Compositional Networks (CCN) Kondor et al. [2018] are most closely related to NGNs, as this is also a local equivariant message passing network. CCN also uses node neighbourhoods and node features that are a representation of the group of permutations of the neighbourhood. CCNs are a special case of NGNs. When in a NGN (1) the node neighbourhood is chosen to be the receptive field of the node, so that the node neighbourhood grows in each layer, and (2) when the edge neighbourhood $\mathcal{G}_{pq}$ is chosen to be the node neighbourhood of $q$, and (3) when the kernel is additionally restricted by the permutation group, rather just its subgroup the automorphism group of the edge neighbourhood, a CCN is recovered. These specific choices make that the feature dimensions grow as the network gets deeper, which can be problematic for large graphs. Furthermore, as the kernel is more restricted, only a subspace of equivariant kernels is used by CCNs.

| Dataset | MUTAG | PTC | PROTEINS | NCI1 | NCI109 | IMDB-B | IMDB-M |
|---|---|---|---|---|---|---|---|
| size | 188 | 344 | 113 | 4110 | 4127 | 1000 | 1500 |
| classes | 2 | 2 | 2 | 2 | 2 | 2 | 3 |
| avg node # | 17.9 | 25.5 | 39.1 | 29.8 | 29.6 | 19.7 | 14 |
| DGCNN [Zhang et al., 2018] | 85.83±1.7 | 58.59±2.5 | 75.54±0.9 | 74.44±0.5 | NA | 70.03±0.9 | 47.83±0.9 |
| PSCN [Niepert et al., 2016][k=10] | 88.95±4.4 | 62.29±5.7 | 75±2.5 | 76.34±1.7 | NA | 71±2.3 | 45.23±2.8 |
| DCNN [Atwood and Towsley, 2016] | NA | NA | 61.29±1.6 | 56.61±1.0 | NA | 49.06±1.4 | 33.49±1.4 |
| ECC [Simonovsky and Komodakis, 2017] | 76.11 | NA | NA | 76.82 | 75.03 | NA | NA |
| DGK [Yanardag and Vishwanathan, 2015] | 87.44±2.7 | 60.08±2.6 | 75.68±0.5 | 80.31±0.5 | 80.32±0.3 | 66.96±0.6 | 44.55±0.5 |
| DiffPool [Ying et al., 2018] | NA | NA | **78.1** | NA | NA | NA | NA |
| CCN [Kondor et al., 2018] | **91.64±7.2** | **70.62±7.0** | NA | 76.27±4.1 | 75.54±3.4 | NA | NA |
| Invariant Graph Networks [Maron et al., 2018] | 83.89±12.95 | 58.53±6.86 | 76.58±5.49 | 74.33±2.71 | 72.82±1.45 | 72.0±5.54 | 48.73±3.41 |
| GIN [Xu et al., 2018] | 89.4±5.6 | 64.6±7.0 | 76.2±2.8 | 82.7±1.7 | NA | **75.1±5.1** | **52.3±2.8** |
| 1-2-3 GNN [Morris et al., 2019] | 86.1 | 60.9 | 75.5 | 76.2 | NA | 74.2 | 49.5 |
| PPGN v1 [Maron et al., 2019] | 90.55±8.7 | 66.17±6.54 | 77.2±4.73 | **83.19±1.11** | 81.84±1.85 | 72.6±4.9 | 50±3.15 |
| PPGN v2 [Maron et al., 2019] | 88.88±7.4 | 64.7±7.46 | 76.39±5.03 | 81.21±2.14 | 81.77±1.26 | 72.2±4.26 | 44.73±7.89 |
| PPGN v2 [Maron et al., 2019] | 89.44±8.05 | 62.94±6.96 | 76.66±5.59 | 80.97±1.91 | 82.23±1.42 | 73±5.77 | 50.46±3.59 |
| Ours (GCN²) | 89.39±1.60 | 66.84±1.79 | 71.71±1.04 | 82.74±1.35 | **83.00 ± 1.89** | 74.80±2.01 | 51.27±1.50 |
| Rank | 5th | 2nd | 11th | 2nd | 1st | 2nd | 2nd |

Table 2: Results on the Graph Classification dataset comparing to other deep learning methods from Yanardag and Vishwanathan [2015].

# 7 Experiments

**Icosahedral MNIST**   In order to experimentally show that our method is equivariant to global symmetries, and increases expressiveness over an invariant message passing network (GCN), we classify MNIST on projected to the icosahedron, as is done in Cohen et al. [2019]. In first column of table 1, we show accuracy when trained and tested on one fixed projection, while in the second column we test the same model on projections that are transformed by a random icosahedral symmetry. NGN outperforms the GCN and the equality of the accuracies shows the model is exactly equivariant. Experimental details can be found in Appendix A.

| Method | Fixed | Sym |
|---|---|---|
| GCN | 96.17 | 96.17 |
| Ours | 98.82 | 98.82 |

Table 1: IcoMNIST results.

**Graph Classification**   We evaluate our model with GCN² message parametrization on a standard set of 8 graph classification benchmarks from Yanardag and Vishwanathan [2015], containing five bioinformatics data sets and three social graphs[5]. We use the 10-fold cross validation method as described by Zhang et al. [2018] and report the best averaged accuracy across the 10-folds, as described by Xu et al. [2018], in table 2. Results from prior work is from Maron et al. [2019]. On most data sets, our local equivariant method performs competitively with global equiviarant methods [Maron et al., 2018, 2019].

In appendix B, we empirically show the expressiveness of our model, as well as the runtime cost.

# 8 Conclusion

In this paper, we have developed a new framework for building neural networks that operate on graphs, which pass messages with kernels that depend on the local graph structure and have features that are sensitive to the direction of flow of information over the graph. We define "natural networks" as neural networks that process data irrespective of how the data is encoded - critically important for graphs, whose typical encoding is highly non-unique - using naturality, a concept from elementary category theory. Local natural graph networks satisfy the naturality constraint with a message passing algorithm, making them scalable. We evaluate one instance of local natural graph networks using a message network on several benchmarks and find competitive results.

## 9 Broader Impact

The broader impact of this work can be analyzed in at least two different ways. Firstly, graph neural networks in general are particularly suited for analyzing human generated data. This makes that powerful graph neural nets can provide tremendous benefit automating common business tasks. On the flip side, much human generated data is privacy sensitive. Therefore, as a research community, we should not solely focus on developing better ways of analyzing such data, but also invest in technologies that help protect the privacy of those generating the data.

Secondly, in this work we used some elementary applied category theory to precisely specify our problem of local equivariant message passing. We believe that applied category theory can and should be used more widely in the machine learning community. Formulating problems in a more general mathematical language makes it easier to connect disparate problem domains and solutions, as well as to communicate more precisely and thus efficiently, accelerating the research process. In the further future, we have hopes that having a better language with which to talk about machine learning problems and to specify models, may make machine learning systems more safe.

## 10 Funding Disclosure

Funding in direct support of this work: Qualcomm Technology, Inc. Additional revenues for Max Welling not used to support this project: part-time employment at the University of Amsterdam.

## Footnotes

*Qualcomm AI Research in an initiative of Qualcomm Technologies, Inc.

[2]Note that the set of node ids may be non-contiguous. This is useful because a graph may arise as a subgraph of another one, in which case we wish to preserve the node ids.

[3]As is common in the category theory literature for functors (see Sec. 5), we overload the $\rho$ symbol. $\rho(G)$ denotes a vector space, while $\rho(\phi)$ denotes a linear map.

[4]In the graph literature, such graphs are also called node/edge rooted graphs.

[5]These experiments were run on QUVA machines.

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
