[Supplementary Material]

# A  Experimental details

**Icosahedral MNIST**   We use node and edge neighbourhoods with $k = 1$. We find the edge neighbourhood isomorphism classes and for each class, the generators of the automorphism group using software package Nauty. The MNIST digit input is a trivial feature, each subsequent feature is a vector feature of the permutation group, except for the last layer, which is again trivial. We find a basis for the kernels statisfying the kernel contstraint using SVD. The parameters linearly combine these basis kernels into the kernel used for the convolution. The trivial baseline uses trivial features throughout, with is equivalent to a simple Graph Convolutional Network. The baseline uses 6 times wider channels, to compensate for the smaller representations.

We did not optimize hyperparameters and have copied the architecture from Cohen et al. [2019]. We use 6 convolutional layers with output multiplicities 8, 16, 16, 23, 23 ,32, 64, with stride 1 at each second layer. After each convolution, we use batchnorm. Subsequently, we average pool over the nodes and use 3 MLP layers with output channels 64, 32 and 10. We use the Adam optimizer with learning rate 1E-3 for 200 epochs. Each training is on one NvidiaV100 GPU with 32GB memory and lasts about 2 hours.

Different from the results in the IcoCNN paper, we are equivariant to full icosahedral symmetry, including mirrors. This harms performance in our task. Further differnt is that we use an icosahedron with 647 nodes, instead of 2.5k nodes, and do not reduce the structure group, so for all non-corner nodes, we use a 7 dimensional representation of $S_7$, rather than a regular 6D representation of $D_6$.

**Graph Classification**   For the graph classification experiments, we again use node and edge neighbourhoods with $k = 1$. This time, we use a GCN message network. At each input of the message network, we add two one-hot vectors indicating $p$ and $q$. The bioinformatics data sets have as initial feature a one-hot encoding of a node class. The others use the vertex degree as initial feature.

We use the 10-fold cross validation method as described by Zhang et al. [2018]. On the second fold, we optimize the hyperparameters. Then for the best hyperparams, we report the averaged accuracy and standard deviation across the 10-folds, as described by Xu et al. [2018]. We train with the Adam optimizer for 1000 epochs on one Nvidia V100 GPU with 32GB memory. The slowest benchmark took 8 hours to train.

We use 6 layers and each message network has two GCN layers. All dimensions in the hidden layers of the message network and between the message networks are either 64 or 256. The learning rate is either 1E-3 or 1E-4. The best model for MUTAG en PTC used 64 channels, for the other datasets we selected 256 channels. For IMDB-BINARY and IMDB-MULTI we selected learning rate 1E-3, for the others 1E-4.

# B  Additional Experiments

**Expressiveness**   Similar to Bouritsas et al [2020], we empirically evaluate the expressiveness of our method. We use a neural network with random weights on a graph and compute a graph embedding by mean-pooling. Then we say that the neural network finds two graphs in a set of graphs to

| Model | Random | Regular | Str. Regular | Isom. |
|---|---|---|---|---|
| GCN | 1 | 6E-8 | 0 | 0 |
| PPGN | 1 | 0.97 | 0 | 6E-8 |
| GCN$^2$ | 1 | 1 | 1 | 6E-8 |

Table 3: Rate of pairs of graphs in set found dissimilar in expressiveness experiment. An ideal method finds only isomorphic graphs not dissimilar.

be different if the graph embeddings differ by an L2 norm of more then a multiple of $\epsilon = 10^{-3}$ of the mean L2 norms of the embeddings of the graphs in the set. The networks is most expressive if it only finds isomorphic graphs to be not different. We test this on (A) a set 100 of random non-isomorphic, non-regular graphs, (B) a set of 100 non-isomorphic regular graphs, (C) a set 15 of non-isomorphic strongly regular graphs (see `http://users.cecs.anu.edu.au/~bdm/data/graphs.html`) and (D) a set of 100 isomorphic graphs, where all graphs have 25 nodes and average of degree 6. We measure average difference rate between pairs of graphs in the sets over 100 different weight initialisations. We compare the simple invariant message passing (GCN), PPGN [Maron et al., 2019], and our GCN$^2$. We see that only our GCN$^2$ can disambiguate between the strongly regular graphs,

showing the expressivity of GCN$^2$. A version of PPGN that uses higher order tensors should also be able to discriminate strongly regular graphs, but at even higher computational cost.

**Runtime Cost**   As an additional experiment we show the runtime cost of one forward-pass of GCN, PPGN and our GCN$^2$. The models have three layers and 32 dimensional activations. For simplicity, we use a square lattice as graph, in which the number of edges is proportional to the number of nodes. In the results below, we observe that GCN$^2$ has indeed a linear scaling and a multiplicative constant about 2x compared to GCN. If the average degree of the graph is higher, this constant may be higher. The global PPGN methods scales superlinearly. Experiments are run on a NVidia GeForce RTX 2080 GPU.

Figure 7: Runtime cost of one forward-pass on square lattices.

## C   Neighbourhood Selection

**Definition C.1.** A *neighbourhood assignment* $\mathcal{N}$, consists of

- a mapping from a graph $G$ and a node $p \in \mathcal{V}(G)$ to node neighbourhood $\mathcal{N}_p(G) \subseteq G$
- a mapping from a graph $G$ and an edge $(p, q) \in \mathcal{V}(G)$ to edge neighbourhood $\mathcal{N}_{pq}(G) \subseteq G$

such that

1. any graph isomorphism $\phi : G \to G'$ restricts to a local node isomorphisms for each node $p \in \mathcal{V}(G)$: $\phi_p := \phi|_{\mathcal{N}_p(G)} : \mathcal{N}_p(G) \to \mathcal{N}_{\phi(p)}(G')$ and to a local edge isomorphism for each edge $(p, q) \in \mathcal{E}(G)$: $\phi_{pq} := \phi|_{\mathcal{N}_{pq}(G)} : \mathcal{N}_{pq}(G) \to \mathcal{N}_{\phi(p)\phi(q)}(G')$

2. for any graph $G$ and edge $(p, q) \in \mathcal{E}(G)$ we have that $\mathcal{N}_p(G) \subseteq \mathcal{N}_{pq}(G) \supseteq \mathcal{N}_q(G)$

3. any local edge isomorphism $\psi : \mathcal{N}_{pq}(G) \to \mathcal{N}_{p'q'}(G')$ restricts to local node isomorphisms: $\psi_0 := \psi|_{\mathcal{N}_p(G)} : \mathcal{N}_p(G) \to \mathcal{N}_{p'}(G')$, $\psi_1 := \psi|_{\mathcal{N}_q(G)} : \mathcal{N}_q(G) \to \mathcal{N}_{q'}(G')$.

The first criterion ensures that global graph isomorphisms translate to local isomorphisms, so that that local naturality implies global naturality. The second and third criteria guarantee that local edge isomorphisms translate into local node isomorphisms, which is necessary for the local naturality criterion to be well-defined. For notational simplicity, we write $G_p := \mathcal{N}_p(G)$ and $G_{pq} := \mathcal{N}_{pq}(G)$.

## D   Proof of global naturality of local NGN kernel

**Theorem 2.** *Let $k$ be a local NGN kernel between node representations $\rho$ and $\rho'$, consisting of for each node neighbourhood $G_{pq}$ a map $k_{pq} : \rho(G_p) \to \rho'(G_q)$ satisfying for any local edge isomorphism $\psi : G_{pq} \to G'_{p'q'}$ that*

$$\rho'(\psi_q) \circ k_{pq} = k_{p'q'} \circ \rho(\psi_p). \tag{7}$$

*Denote by $\hat{\rho}$ and $\hat{\rho}'$ the global graph representations induced by local node representations $\rho$ and $\rho'$. Then the layer*

$$K_G(v)_q = \sum_{(p,q)\in\mathcal{E}(G)} k_{pq}(v_p) \tag{8}$$

*satisfies the global NGN naturality condition, for any global graph isomorphism $\phi : G \to G'$*

$$\hat{\rho}'(\phi) \circ K_G = K_{G'} \circ \hat{\rho}(\phi). \tag{9}$$

*Proof.* We need to show that for any feature $v \in \hat{\rho}(G)$, that $\hat{\rho}'(\phi)(K_G(v)) = K_{G'}(\hat{\rho}(\phi)(v)) \in \hat{\rho}'(G')$, which we do by showing the node features are equal at each $q' \in \mathcal{V}(G')$. Denote $\phi_p$ and $\phi_q$ as

the restriction of graph isomorphism $\phi : G \to G'$ to the node neighbourhoods of $p$ and $q$. Let $p' = \phi(p), q' = \phi(q)$. Then we have that

$$\hat{\rho}'(\phi)(K_G(v))_{q'} = \rho'(\phi_q)(K_G(v)_q)$$

$$= \rho'(\phi_q)\left( \sum_{(p,q)\in\mathcal{E}(G)} k_{pq}(v_p) \right)$$

$$= \sum_{(p,q)\in\mathcal{E}(G)} \rho'(\phi_q)(k_{pq}(v_p))$$

$$= \sum_{(p,q)\in\mathcal{E}(G)} k_{p'q'}(\rho(\phi_p)(v_p))$$

$$= \sum_{(p,q)\in\mathcal{E}(G)} k_{p'q'}(\hat{\rho}(\phi)(v)_{p'})$$

$$= \sum_{(p',q')\in\mathcal{E}(G')} k_{p'q'}(\hat{\rho}(\phi)(v)_{p'})$$

$$= K_{G'}(\hat{\rho}(\phi)(v))_{q'}.$$

where in the third line we use linearity of $\rho'$, in the fourth line we recognise that $\phi$ restricts to local edge isomorphism $\phi_{pq}$ and apply the constraint on the local NGN kernel and in the fifth line we use the bijection between $\mathcal{E}(G)$ and $\mathcal{E}(G')$. $\qquad\square$

## E   Message Network gives Local NGN Kernel

To define the message network, we first need to define node features $\rho, \rho'$ and edge features $\tau, \tau'$, completely analog to how node features are defined. Furthermore, we need for each edge $(p, q)$ embedding map $\alpha_{pq} : \rho(G_p) \to \tau(G_{pq})$ and projection map $\beta_{pq} : \tau'(G_{pq}) \to \rho'(G_q)$. These should satisfy that for any edge isomorphism $\psi : G_{pq} \to G'_{p'q'}$, $\alpha_{p'q'} \circ \rho(\psi_p) = \tau(\psi) \circ \alpha_{pq}$ and $\beta_{p'q'} \circ \tau'(\psi) = \rho'(\psi_q) \circ \beta_{pq}$, meaning that isomorphisms commute with embeddings and projections. For each edge $(p, q)$ the adjacency matrix can be encoded as an edge feature $\tau$ as matrix $A_{pq} \in \tau_A(G_{pq})$.

When all representations are tensor products of the standard representation of the permutation group, we can use a single message network $\Psi$ taking as input the embedding of the input node feature $\alpha_{pq}(v_p)$ and the adjacency matrix $A_{pq}$ and outputting an output edge feature $\tau'(G_{pq})$. When $\Psi$ is an equivariant graph network, we have that $\sigma\Psi(v, A) = \Psi(\sigma v, \sigma A)$ for any permutation $\sigma$ in the appropriate permutation representation. The local NGN kernel is then defined as $k_{pq}(v_p) = \beta_{pq}(\Psi(\alpha_{pq}(v_p), A_{pq}))$.

Then this kernel satisfies the local NGN naturality for any edge isomorphism $\psi : G_{pq} \to G'_{p'q'}$ (Eq. 4):

$$k_{p'q'}(\rho(\psi_p)(v)) = \beta_{p'q'}(\Psi(\alpha_{p'q'}(\rho(\psi_p)(v_p)), A_{p'q'}))$$

$$= \beta_{p'q'}(\Psi(\tau(\psi)(\alpha_{pq}(v_p)), A_{p'q'}))$$

$$= \beta_{p'q'}(\Psi(\tau(\psi)(\alpha_{pq}(v_p)), \tau(\psi)(A_{pq})))$$

$$= \beta_{p'q'}(\tau'(\psi)(\Psi(\alpha_{pq}(v_p), (A_{pq}))))$$

$$= \rho'(\psi_q)\beta_{pq}(\Psi(\alpha_{pq}(v_p), (A_{pq}))))$$

$$= \rho'(\psi_q)(k_{pq}(v_p))$$

where in the second line we used the commutation of $\alpha$, in the third line we use that $A_{p'q'} = \tau(\psi)(A_{pq})$ as an immediate consequence of the fact that $\psi$ is an edge neighbourhood isomorphism, in the fourth line the equivariance of $\Psi$, in the fifth line the commutation of $\beta$,

Figure 8: Node and edge neighbourhood on a triangular tiling.

# F    Reduction to Group & Manifold Gauge Equivariance

The two dimensional plane has several regular tilings. These are graphs with a global symmetry that maps transitively between all faces, edges and nodes of the graph. For such a tiling with symmetry group $G \rtimes T$, for some point group $G$ and translation group $T$, we can show that when the neighbourhood sizes and representations are chosen appropriately, the natural graph network is equivalent to a Group Equivariant CNN of group $G$ Cohen and Welling [2016].

For sufficiently large node neighbourhoods, the node automorphisms equal the point group $G$ of the lattice, thus, from any representation $\rho_G$ of $G$ with representation space $V$, we can build a node feature $\rho$ with for each node $p$, $\rho(G_p) = V$ and in which all node isomorphisms $\psi$ have $\rho(\psi) = \rho_G(g)$ for some group element $g \in G$. The way to construct this, is to pick one reference node $p$, make an identification of the automorphism group $\mathrm{Aut}(G_p)$ with $G$ and then for all isomorphic nodes $p'$ pick one isomorphism $\psi : G_p \rightarrow G'_{p'}$ with $\rho(\psi) = \mathrm{id}_V$. The functor axioms then fully specify $\rho$.

Now, as an example consider one of the tilings of the plane, the triangular tiling. As shown in figure 8, the node neighbourhood has as automorphism group the dihedral group of order 6, $D_6$, so we can use features with reduced structure group $D_6$. The kernel $k_{pq}$ is constrained by one automorphism, which mirrors along the edge. A Natural Graph Network on these reduced features is exactly equivalent to HexaConv [Hoogeboom et al., 2018]. Furthermore, the convolution is exactly equivalent to the Icosahedral gauge equivariant CNN [Cohen et al., 2019] on all edges that do not contain a corner of the icosahedron. A similar equivalence can be made for the square tiling and a conventional $D_4$ planar group equivariant CNN [Cohen and Welling, 2016] and a gauge equivariant CNN on the surface of a cube.