[Reviews · NeurIPS 2020]

Review 1

Summary and Contributions: This paper considers the problem of designing graph neural networks that are more expressive than ordinary message passing networks (by treating the local neighborhood as more than just a set of neighbors), but still efficient to use (i.e. without having nonlinear blowup as the graph size increases). The authors propose a framework for graph neural networks based on local isomorphisms, for which the node features transform (equivariantly) based on changes to local structure and do not change based on nonlocal changes to the global structure, and for which the message-passing transformations transform in a similar way. As a concrete instantiation of this framework, they propose GCN^2, which represents each node's features not as a single feature vector, but as a map from that node's neighbors to features (implemented as a vector whose elements permute when the node's neighbors are permuted), and performs message passing by running a local GCN within the neighborhood of each edge to propagate features from one node's neighborhood to another node's neighborhood. The authors also briefly introduce a formalization of this idea in terms of category theory, and argue that their framework can be represented simply by a set of category-theoretic constraints in that formalization.

Strengths: The idea of representing nodes with fixed-size neighborhoods, and then constraining the message-passing structure such that it obeys the necessary equivariance constraints, seems interesting. In addition, the framework the authors propose appears to be quite general, and might lead to future work regarding more powerful graph neural networks. I cannot speak directly to novelty, however, because I am not very familiar with the prior work in this space. The category-theory interpretation of the idea seems particularly interesting, and I think provides context for the constraints the authors describe, as well as a good way of thinking about how to design new graph neural network models in general. The paper provides experimental results across a reasonable set of benchmark tasks.

Weaknesses: The main weakness, in my opinion, is the presentation of the experimental results. The architecture is motivated primarily based on efficiency and asymptotic complexity, with the goal being to construct a model that is more powerful but still efficient to train and scalable to large graphs. However, the authors do not include any measurements of runtime of their model vs. baselines, which makes it difficult to judge whether their model is actually better in this respect, and if so, how much. Additionally, the authors claim that "On most data sets, our local equivariant method performs competitively with global equiviarant [sic] methods", but this is not obvious from the results they provide (as far as I can tell, their model seems to perform roughly middle-of-the-pack for most tasks) and seems like a difficult claim to quantify. I also don't understand why the non-deep-learning approaches are presented separately from the ones they compare against directly. I was hoping for some sort of Pareto frontier graph comparing runtime with accuracy, and showing that their method is better than others for some computational or memory budget. Another thing that wasn't clear to me: how powerful is this new model in relation to prior work? In particular, does using another GCN to parameterize the message-passing of GCN^2 really lead to a model that is more expressive than GCN is alone, at least in theory? And how close is that expressivity to some of the "monolithic" approaches the authors describe in the introduction?

Correctness: As far as I can tell, the high level claims of the paper are correct. In particular, it seems true that, under the constraints they describe, models that follow their framework will be globally invariant and thus well-formed as graph neural networks. There are a few statements that seem slightly sloppy, which I have noted in the "additional feedback" section below. It is difficult to evaluate the experiments themselves, due to a small amount of details being provided, but nothing seems particularly problematic. However, their claim that their method is competitive seems very vague, and I'm not sure it is supported well by their results.

Clarity: The motivation and background sections seemed clear and well written. However, sections 3 and 4 were hard for me to follow, in some cases asserting things were true that didn't seem obvious to me, and going into specific details without motivating why they were chosen. I liked section 5, which provided an abstract view of the idea in terms of category theory concepts, but it seemed presented almost as an afterthought. Perhaps it would be better to put this section first, using it to give a birds-eye view of the different pieces of the model, and then refer back to it when going into details. Regarding the category-theory interpretation, I also wonder whether it would be possible to simplify the idea by introducing an additional category of node neighborhoods and a functor mapping from node neighborhoods to node feature vector spaces, and from isomorphisms to permutation group actions.

Relation to Prior Work: The prior work section only goes into specific details regarding CCN and not the other types of more expressive equivariant GNNs. I'm not knowledgeable enough regarding prior work to know if this is a reasonable amount of detail, or if there was anything important missed.

Reproducibility: No

Additional Feedback: Specific notes: Line 132, "local symmetries form a superset of the global symmetries": I'm not sure what this means. Local and global symmetries are different types of object, right? Is this just intended as meaning that global symmetries restrict to local ones? Line 147: "it is necessary that the feature vector ... transforms ... rather than remain invariant" Is this actually true, and if so, what is the justification? It seems to me like using a non-transforming feature vector with a powerful edge kernel could still be possible? Line 176: "For an isomorphism between different edges, this implies that the kernel is shared." I don't understand what this is saying. Is this just a restatement of (4)? I initially interpreted "shared" as meaning "identical" but do you just mean "transforms equivariantly" or something similar? Also, should this be an isomorphism of edge *neighborhoods*? Figure 6: This figure was hard for me to understand. Firstly, is there a color missing in the top left vector? Seems like one of the colors is duplicated. Second, it's very non-obvious to me what the coloring of the graphs represents. Is it that dark blue and light blue (which I found a bit hard to distinguish) are the local and global node neighborhoods? And the reason node labels don't move for the automorphism is because it is an automorphism, whereas the isomorphism permutes the labels? Does the fact that there are two neighbors of 3 in the third diagram have anything to do with isomorphism vs automorphism? And similarly, the position of the other node in the first two diagrams? In general having a much more detailed figure caption would help here. Line 204: What is "a graph feature"? This notation has never been defined. I also don't understand why the claim about embeddability is true. Line 241: The definition of a morphism is incorrect, and should go to G(X). Minor comments: Figure 1 caption: typo "isomorphisms" Line 172: should "and" be "to"? Line 246: typo "space space" ===== Edit: after rebuttal ===== Thank you for the additional experimental results provided in the author response, which seem to demonstrate that, at least on some simple tasks, the GCN^2 model is more powerful than a regular GCN and also faster than the PPGN model. I still think the paper could be greatly improved by adding similar information about runtime to the experiments in table 2. For instance, since the GCN^2 model is more efficient than other approaches, one might hope that it would outperform other methods given a fixed computational budget. As it is now, it seems that the GCN^2 model is demonstrably powerful on artificial tasks but not necessarily that useful in practical settings. I have increased my review to 6 in light of the additional results.


Review 2

Summary and Contributions: This work proposes a novel graph network that assigns different kernels to different edges based on the graph structure. The proposed method can make graph networks achieve local and global graph isomorphisms. Moreover, a theoretical analysis of the method is provided.

Strengths: This work is well-written. The authors propose a novel kind of message passing network that assigns weights to edges depending on the graph structure. The correctness of the model is guaranteed by theoretical analysis. Existing GCNs can easily implement the proposed model.

Weaknesses: My concerns are listed as following: Firstly, some details are missing, which makes the method confusing. For example, what is the restricted operation in Figure 7? Why do the authors change node positions of the four graphs in Figure 7? Secondly, the authors claim that no invariant message-passing network can discriminate between the two graphs in Figure 2. I wonder whether the proposed method can do this. Can the authors provide in-depth analysis? Thirdly, the experimental part is weak. I would like to see the experiments where assigning different weights to different edges is beneficial. Moreover, the proposed method only achieves state-of-the-art performance in 1 out of 8 benchmarks.

Correctness: This work is theoretically correct.

Clarity: This work is well-written.

Relation to Prior Work: This work discusses the difference between previous contributions.

Reproducibility: No

Additional Feedback: The authors need to modify Section 4 and add more details about their message passing. It will be beneficial to add extra experiments to indicate the benefit of assigning different weights to different edges. The authors didn't address my concerns about the weak experiments, so I will not change the overall score for this submission.


Review 3

Summary and Contributions: It has been pointed out in the literature, both by informal and theoretical arguments, that message passing neural networks are inherently limited in their expressive capacity. In particular, they are invariant to node re-labelings, and thus lose topological information about the graph. With the goal of designing more powerful graph networks, many methods have been proposed. In particular, two methods discussed in the paper include ones proposed by Kondor et al and Maron et al, which are strictly more expressive than MPNNs, being equivariant to node relabelings. However, these methods, treat the graph as a single linear structure and thus have higher attendant computational cost.The main contribution of this paper is to design a graph network that is maximally expressive, equivariant to global node relableings, while only relying on local computation. The main idea of the paper is that restricting a global node relabeling to a local neighborhood implies an isomorphism between the neighborhoods. Any message passing scheme should operate on such neighborhoods in a similar manner, giving a prescription for weight sharing. Furthermore, the kernel used must also satisfy an equivariance constraint with respect to the symmetry group of the neighborhood. Using the local computation of MPNNs as a starting point, weights of the kernel are made to depend on the structure of the graph (and thus can vary per graph and per edge). However not all such kernels will be equivariant. The paper then derives such kernels. In particular, it only works with a local context for a particular edge. If any other edge has a similar local structure around it, then they will share weights. The local neighborhood symmetries are required to be a superset of the global symmetries, so that equivariance to local symmetries will imply equivariance to global symmetry. The equivariance constraints are derived simply by using the fact that a global isomorphism \phi will imply a corresponding local isomorphism \phi' in a particular neighborhood by restriction. Similar symmetries can be defined for nodes and substructures as well. The paper is most related to two papers: - Gauge equivariant models: In the sense of also using local symmetry. But since each edge can have different kernels, we don't work with a single gauge, but rather a symmetry groupoid. - Covariant Compositional Networks: The similarity can be seen clearly in figure (4), where "natural" networks can be seen to be essentially the same as CCN but without the "tensor promotion" step described in the CCN paper. The tensor promotion step makes CCN global, and thus computationally expensive. The paper then also uses simple category theory to unify the various varieties of equivariant networks (on graphs, homogeneous spaces, manifolds) into a single framework. The details are mostly confined to the appendix.

Strengths: The paper makes significant and novel contributions along two directions: -- Derives graph networks that are maximally expressive and yet rely only on local computation. The work is a generalization on the work on equivariant graph networks and message passing networks. It also provides a fresh prespective in the space after guage equivariant models. -- Provides a category theory based language to talk about different flavours of equivariant networks in a common framework. I think the language described would be quite influential in understanding and deriving new equivariant networks in the future.

Weaknesses: The experimental results are a little disappointing, even surprising. First: The tasks considered are not exactly involved (except for the last three columns of table 2), but still fair, as they adequately compare to related methods. Second: Intuitively, it is clear that the method described should at least do as well as other models (which is not always borne out in the results). This aspect is surprising and I wonder what part of the implementation would limit the results (have the authors tried to understand this?). I would expect the proposed method to have stronger results than any of the equivariant graph networks consistently. Third: Because of the use of local symmetries, I would expect the implementation to be quite involved. It would be nice to also have timing for the results if possible, compared to one equivariant network and MPNN. While this suggestion might seem lazy. I am making this so because I am curious how the proposed method differs in complexity with the other, arguably simpler, methods.

Correctness: Yes.

Clarity: The paper is very well written. Section 5 can be a little better written. It strikes as foisted out of the blue and without much motivation. It is clear that the section makes an interesting contribution on its own (even if most of the meat is transported to the appendix), and it would be good to have this borne out in the writing.

Relation to Prior Work: Prior work is clearly described and the paper placed appropriately in context.

Reproducibility: Yes

Additional Feedback:


Review 4

Summary and Contributions: The paper's goal is to (a) improve the expressive capacity of MPNNs, especially wrt loss of structural information during message-passing (eg inability to distinguish certain non-isomorphic graphs), while (b) maintaining the efficiency advantages standard MPNN approaches have over more expressive models such as Kondor's and Lipman's. The general idea is rather than enforcing global permutation equivariance, all that is needed is to enforce local equivariance in node/edge nieghborhoods. This can be accomplished by parameterizing the message-passing computations differently for each distinct local neighborhood isomorphism. In other words, standard MPNNs allow non-isomorphic neighborhoods to be embedded into the same representation, but by using different kernels for non-isomorphic neighborhoods, the authors' proposed NGN approach has the ability to avoid this loss of structural information. Because heterogeneous graphs may not have many local isomorphic neighborhoods, a naive NGN would not exploit much parameter sharing because each neighborhood would have a different kernel. This would not promote efficient learning and generalization. However the authors show that by including local neighborhood features on the nodes and edges, and using a shared kernel, the kernel can be aware of local non-isomorphism (in the extreme, it could learn a lookup table of sub-functions specific to different neighborhoods), while still being shared globally. As a caveat, the above summary is my best guess of what the paper has done, but I'm not that confident I understood it fully.

Strengths: The paper has recognized an important problem in the literature and offered a strong theoretical solution. If my understanding was correct, it can help organize MPNNs, the Kondor and Lipman bodies of work, and also several directions the authors didn't cite, which involve different node labeling schemes to break symmetries and overcome weaknesses GNNs have in distinguishing isomorphic graphs.

Weaknesses: The paper has two weaknesses: 1. The value of the central theoretical contribution is undermined by the overly technical nature of the presentation. If my interpretation was correct, the paper's idea should have been much easier to get across (and if my interpretation was incorrect, that's not good news either...). For example, the concept of bijections over node orderings is raised multiple times across different pages (lines 60, 102, and 159) in different superficial contexts, though they're all connected. Relatedly, the \phi mapping is defined 5 times, on lines 102, 135, 165, 173, and 249, twice as a global node isomorphism, once as a local node isomorphism, and twice as a local edge isomorphism. Then on line 136 \phi' is defined as a local isomorphism. Permutations and gauges are also defined similarly here and there. I was jumping back and forth between pages before I gave up on the details, and recognized I could only get the general idea. I also found the the discussion of gauges and category theory unnecessary. The category theory is confined to a dedicated section, which is fine, but why mention it in the Abstract? Could it be in the Supp Mat? As a broad suggestion, I think it'd be more effective if the most technical parts of the theory were put in a single, dense section, with equation environments rather than inline equations/notation, and walked through carefully and concisely, then referenced heavily later. 2. The empirical results are not particularly strong compared to baselines, and I suspect if other approaches, such as Position-aware GNNs (You 2019), Relational Pooling (Murphy 2019), “Random Features Strengthen Graph Neural Networks” (Sato 2020) were compared, they may fare as well or better, because my sense is that part of what they're doing underneath is approximating what NGN is doing more explicitly. These papers were not cited (and there are a number of other papers which are similar and relevant), so it's unclear to me whether the authors are aware of them.

Correctness: Correct

Clarity: I found the paper very difficult to read. I kept getting lost in the terminology (local vs global, symmetries/equivariance/invariance/isomorphism/automorphism), as mentioned in my Weaknesses comments above. The figures (especially 1, 3-7) were quite unintuitive to me. Some figures have single sentence captions. Figure 1 doesn't describe the (a)/(b) sub-figures separately. I'm not sure what the node and vector element colors were supposed in indicate. I'm not sure what the permuted indices, and layout reflections indicate. For example, in figure 1 what do the colored edge arrows signify, or the dotted crisscrossed arrows? In figure 3 why are the top graphs reindexed and reflected (over either the horizontal or vertical axis)? In figure 7, why does the Embed step include a permutation of some of the nodes, and what do the edge arrows between 1->3 and then 3->1 mean? The fact that the figures are all obviously closely related, not easy to interpret, and spread over 6 pages highlights the disorganized nature of the presentation.

Relation to Prior Work: The relation to the more theoretical work, eg Kondor/Lipman's work on improving over WL, the equivariance work in Welling's group, etc, are well cited. The more applied work in the GNN community isn't as well cited, and I suspect it has close connections to this.

Reproducibility: Yes

Additional Feedback: I want this paper to be successful because I think there's substantial value in it. However I can't recommend it be accepted, as is, because I think its presentation isn't clear enough, its connection to the more applied literature is insufficiently explored, and the empirical results are not very compelling. My suggestion would be to reorganize the writing so that the technical content is confined to 1.5-2 pages, much more intuition is provided using better figures and simple examples, and more explanation of how the theory introduced here may help unify the less principled work in the GNN community that aims to overcome weaknesses GNNs have with isomorphic graphs.

[Author Response · NeurIPS 2020]

We would like to thank all reviewers for taking the time to provide our work with extensive and constructive feedback, which will be very useful for improving our work for the final version.

- **R1 and R2: expressiveness** R1 and R2 requested a demonstration that the resulting $GCN^2$ method is indeed more expressive than a GCN. Contrary to the global equivariant methods, our method can not implement a Weisfeiler-Lehman test, making theoretical analysis more difficult. Instead, we use an empirical evaluation, similar to [1]. We use a neural network with random weights on a graph and compute a graph embedding by mean-pooling. Then we say that the neural network finds two graphs in a set of graphs to be different if the graph embeddings differ by an L2 norm of more then a multiple of $\epsilon = 10^{-3}$ of the mean L2 norms of the embeddings of the graphs in the set. The networks is most expressive if it only finds isomorphic graphs to be not different. We test this on (A) a set 100 of random non-isomorphic, non-regular graphs, (B) a set of 100 non-isomorphic regular graphs, (C) a set 15 of non-isomorphic strongly regular graphs (see http://users.cecs.anu.edu.au/~bdm/data/graphs.html) and (D) a set of 100 isomorphic graphs, where all graphs have 25 nodes and average of degree 6. We measure average difference rate between pairs of graphs in the sets over 100 different weight initialisations. We compare the simple invariant message passing (GCN), PPGN [2], and our $GCN^2$. We see that only our $GCN^2$ can disambiguate between the strongly regular graphs, showing the expressivity of $GCN^2$. A version of PPGN that uses higher order tensors should also be able to discriminate strongly regular graphs, but at even higher computational cost.

- **R1 and R3: runtime cost?** As an additional experiment we show the runtime cost of one forward-pass of GCN, PPGN and our $GCN^2$. The models have three layers and 32 dimensional activations. For simplicity, we use a square lattice as graph, in which the number of edges is proportional to the number of nodes. In the results below, we observe that $GCN^2$ has indeed a linear scaling and a multiplicative constant about 2x compared to GCN. If the average degree of the graph is higher, this constant may be higher. The global PPGN methods scales superlinearly. We will add more extensive experiments to the final version.

- **R1, R2 and R6: figures unclear.** We will add extensive captions to the figures to clarify.

- **R1: local symmetries superset?** We indeed mean that global syms restrict to local syms and will clarify.

- **R1: equivariance necessary for expressiveness?** In its simplest linear form, an invariant message passing network equals a GCN with limited expressivity, making more general equivariance necessary for equivariance. We agree with R1 that for more general non-linear kernels, such a general statement does not hold. We will clarify this.

- **R1: kernel shared?** Indeed, we mean that the parameters are shared, but the resulting kernels differ by a basis transformation. This is implied by Eq. (4), when the isomorphism is between different edges neighbourhoods. We used edge isoms and edge neighbourhood isoms interchangeably. We will clarify these points.

- **R1: graph feature?** By graph feature, we mean some feature over an entire graph, e.g. the stacking of all node features for a conventional GCN or the matrix feature of [2]. The idea of Sec 4 and Fig 7 is to embed a node feature (in the vector representation) into a feature of the edge neighbourhood and then to re-interpret the the edge feature of one edge as an invariant GCN-like graph feature over the edge neighbourhood graph, having a scalar feature at each node of the edge neighbourhood. Please see App. C for more details. We'll clarify this in Sec 4.

- **R1: corrections.** Thank you for finding the typos. We will correct them.

- **R2: discriminate Fig 2?** We ran the experiment mentioned above additionally on the graphs of Fig 2 and found that a conventional GCN always returns the same embedding, while our $GCN^2$ is able to discriminate.

- **R6: overloading of symbols.** Thank you for pointing out the fact that our re-use of the symbol $\phi$ for distinct related concepts is unclear. We will fix that.

- **R6: technicality.** We think that the use of gauges are a necessity to precisely describe the feature spaces and equivariant maps. Also, we agree with R1 and R3 that the categorical perspective is a useful ways of thinking about these models. However, we agree with R6 that besides the technical sections, we should explain better intuitively how the resulting model works. We will fix this in the final version.

- **R6: references.** We thank R6 for providing three interesting and relevant references. We will relate our model to these in the final version.

| Model | Random | Regular | Str. Regular | Isom. |
|---|---|---|---|---|
| GCN | 1 | 6E-8 | 0 | 0 |
| PPGN | 1 | 0.97 | 0 | 6E-8 |
| $GCN^2$ | 1 | 1 | 1 | 6E-8 |

**Table 1:** Rate of pairs of graphs in set found dissimilar in expressiveness experiment. An ideal method finds only isomorphic graphs not dissimilar.

**Figure 1:** Runtime cost of one forward-pass on square lattices.

[1] Giorgos Bouritsas et al. Improving graph neural network expressivity via subgraph isomorphism counting. 2020.

[2] Haggai Maron, Heli Ben-Hamu, Hadar Serviansky, and Yaron Lipman. Provably powerful graph networks. 2019.


[Meta-Review · NeurIPS 2020]

While I think all reviewers and myself included appreciate the value of the work, there are a few lingering concerns around the clarity of the presentation and potentially slightly weak empirical evidence. In particular there are questions about comparison with additional work (as pointed out by L6). I think the value outweighs the negatives. However I urge the authors to incorporate as much as they can from the feedback that they got, and the details mention in the rebuttal, in order to maximize the impact this work can have within the community!